# Adaptive Federated Learning Defences via Trust-Aware Deep Q-Networks

## Abstract

Federated learning is vulnerable to poisoning and backdoor attacks under partial observability. We formulate defence as a partially observable sequential decision problem and introduce a trust-aware Deep Q-Network that integrates multi-signal evidence into client trust updates while optimizing a long-horizon robustness–accuracy objective. On CIFAR-10, we (i) establish a baseline showing steadily improving accuracy, (ii) show through a Dirichlet sweep that increased client overlap consistently improves accuracy and reduces ASR with stable detection, and (iii) demonstrate in a signal-budget study that accuracy remains steady while ASR increases and ROC-AUC declines as observability is reduced, which highlights that sequential belief updates mitigate weaker signals. Finally, a comparison with random, linear-Q, and policy gradient controllers confirms that DQN achieves the best robustness–accuracy trade-off.

## 1 Introduction

Federated learning enables collaborative model training across distributed participants without centralizing sensitive data, making it attractive for privacy-critical applications from healthcare to mobile devices (McMahan et al., 2023). However, this decentralized paradigm introduces fundamental security vulnerabilities: malicious participants can inject model poisoning attacks that degrade global performance or backdoor attacks that embed hidden malicious behaviors triggered by specific inputs (Bagdasaryan & Shmatikov, 2021; Bhagoji et al., 2019).

### 1.1 Moving Beyond Static Defenses

Current FL defense mechanisms face two critical limitations that constrain their real-world effectiveness:

**Partial Observability.** Modern FL deployments often employ secure aggregation protocols that prevent servers from accessing raw client updates. Defenders must infer malicious behavior from limited indirect signals, creating an inherently uncertain detection problem where ground truth client honesty remains hidden.

**Adaptive Adversaries.** Sophisticated attackers continuously evolve their strategies, adjusting attack magnitudes, directions, and timing to evade static detection rules. Fixed threshold-based defenses become ineffective as attackers learn to mimic benign client behavior patterns.

Classical robust aggregation methods (coordinate-wise median, trimmed mean, Krum (Blanchard et al., 2017)) provide theoretical guarantees only under restrictive assumptions such as majority honesty or statistical independence, that rarely hold in practice. Trust-based approaches like FLTrust (Cao et al., 2022) and FLARE (Wang et al., 2022) improve robustness through reference datasets or consistency checks, but still rely on static similarity metrics and per-round decisions that adaptive attackers can systematically exploit.

### 1.2 Our Approach: Sequential Decision Making Under Uncertainty

We take a fundamentally different perspective modeling federated learning defense as a partially observable sequential decision problem. The server must (i) infer hidden client trustworthiness

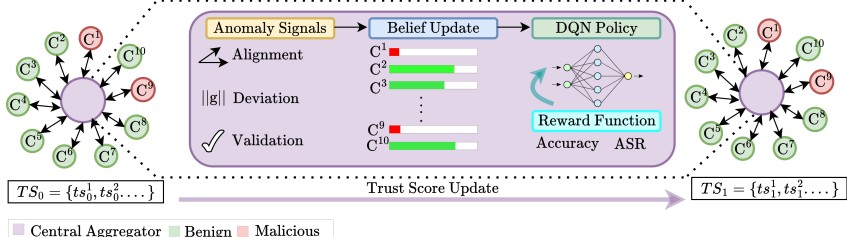

Figure 1: **Trust-aware RL defence under partial observability.** Client updates are converted to anomaly signals: alignment, $\|g\|$ magnitude deviation, and validation impact. This feeds to a Bayesian belief state update module. The DQN policy then acts on these beliefs by choosing, for each client, to *increase*, *decrease*, or *hold* its trust score, producing $TS_{t+1}$. These trust scores parameterize aggregation (reweighting/gating of updates) to form the global model update, and a reward based on accuracy and ASR trains the policy over rounds under secure aggregation.

from noisy, incomplete observations, (ii) accumulate evidence across multiple training rounds, (iii) adaptively decide how to weight or filter client contributions and (iv) balance immediate accuracy with long-term robustness

This perspective naturally leads to a reinforcement learning formulation where an intelligent agent learns optimal defense policies through interaction with the federated environment.

### 1.3 CONTRIBUTIONS

Our work makes four key contributions to augment a DQN-based dynamic defence in FL:

1. **POMDP Formulation**: We introduce the first principled framework that casts FL defense as a partially observable Markov decision process, treating client trust as latent state and server aggregation as sequential decision-making under uncertainty.
2. **Multi-Signal Bayesian Trust Tracking**: We design a sophisticated evidence fusion pipeline combining directional alignment, magnitude deviation, and validation impact with Bayesian belief updates, enabling dynamic trust scores that accumulate evidence across rounds rather than making myopic per-round decisions.
3. **Adaptive RL Aggregator**: We develop a trust-aware Deep Q-Network that learns to optimize robustness-accuracy trade-offs by adaptively filtering and reweighting client updates, substantially outperforming static robust aggregators and prior RL approaches.

Together, these contributions establish reinforcement learning under partial observability as both a principled and practical foundation for robust, adaptive federated learning that can defend against sophisticated adversaries in realistic deployment scenarios.

## 2 RELATED WORK

### 2.1 ATTACKS ON FEDERATED LEARNING

Federated Learning (FL) offers privacy by keeping data local, but its distributed nature enlarges the attack surface. Early poisoning attacks relied on simple manipulations such as sign-flipping or scaling to disrupt aggregation (Baruch et al., 2019). Fang et al. (Fang et al., 2020) later formalized poisoning as an optimization problem, enabling adversaries to craft updates that strategically maximize deviation. More recent work, such as PoisonedFL (Xie et al., 2025), shows that enforcing multi-round consistency allows malicious clients to persistently evade state-of-the-art defenses, while reconnecting malicious clients (RMCs) can exploit FL's open connectivity to re-enter with new strategies (Szeląg et al., 2025). These advances highlight that poisoning resilience requires temporal memory and adaptivity.

Backdoor attacks pose an even stealthier threat, embedding hidden triggers while maintaining high clean accuracy. Blind backdoors compromise the training loss to inject triggers without modifying

data, evading sanitization defenses (Bagdasaryan & Shmatikov, 2021) and that even a few malicious clients can implant backdoors. Distributed Backdoor Attacks (DBA) (Liu et al., 2025) further exploit federation by splitting triggers across clients. These works demonstrate that per-round anomaly detection is insufficient against coordinated, temporally distributed backdoors.

## 2.2 DEFENSES AGAINST MALICIOUS CLIENTS

The first line of defense was Byzantine-robust aggregation, including Krum (Blanchard et al., 2017), coordinate-wise median, and trimmed mean (Yin et al., 2021). While these methods provide theoretical guarantees under bounded adversaries and IID data, they fail in realistic FL with non-IID distributions, collusion, or strategically crafted updates (Baruch et al., 2019).

Trust-based defenses attempt to assign reputations: FLTrust (Cao et al., 2022) anchors updates to a clean server dataset, while FLARE (Wang et al., 2022) clusters latent representations. However, these methods remain myopic, treating each round independently. Identity-based cryptographic schemes (Szeląg et al., 2025) offer stronger guarantees against reconnecting adversaries, but still lack mechanisms for sequential trust tracking.

Finally, learning-based defenses explore reinforcement learning for adaptive aggregation. Yet most assume full observability of client updates, modeling the problem as a Markov Decision Process. This is misaligned with realistic FL, where secure aggregation creates partial observability. Our work addresses this gap by framing defense as a Partially Observable MDP (POMDP), enabling belief tracking of hidden client trustworthiness and adaptive policies that reason across rounds.

## 3 METHODOLOGY

We formulate federated learning defence against adversarial clients as a sequential decision-making problem under uncertainty. The server receives client updates at each round but cannot directly observe whether they are benign or malicious. Robust aggregation therefore requires inferring hidden trustworthiness from indirect signals, accumulating this information across rounds, and adaptively deciding which clients to prioritize. We cast this setting as a partially observable Markov decision process (POMDP) and design a trust-aware reinforcement learning framework that combines three elements: anomaly-based evidence extraction, Bayesian belief tracking of trust, and a Deep Q-Network (DQN) policy that learns adaptive aggregation strategies. Figure 1 provides an overview.

### 3.1 FEDERATED LEARNING WITH MALICIOUS CLIENTS

We consider a cross-device FL setting with $N$ clients jointly training a global model. At round $t$, each client $i$ computes a local update $w_i^t$ on its private data and sends it to the server. A fraction $\alpha$ of these clients may be malicious, submitting poisoned updates intended to degrade accuracy or implant backdoors. The global model evolves as

$$w^{t+1} = \mathcal{A}(\{w_i^t\}),$$

where $\mathcal{A}$ denotes the server's aggregation rule. The challenge is to design $\mathcal{A}$ so that benign updates dominate while malicious contributions are suppressed, even though the server never directly observes client honesty.

### 3.2 TRUST-WEIGHTED AGGREGATION

To move beyond uniform averaging, we assign each client a dynamic trust score $TS^t(i) \in [0, 1]$ that modulates its weight:

$$w^{t+1} = \sum_{i=1}^{N} \frac{n_i}{n} TS^t(i) w_i^t, \qquad n = \sum_{i=1}^{N} n_i,$$

where $n_i$ is the number of training samples at client $i$. Initially $TS^0(i) = 1$ for all clients, and scores are adjusted across rounds based on anomaly evidence. This formulation generalizes prior trust-based defences as demonstrated in FLTrust(Cao et al., 2022) and FLARE(Wang et al., 2022) by allowing trust to evolve adaptively in response to observed behaviour rather than remaining fixed.

---

**Algorithm 1** Round-level trust-aware DQN defence

---

1: **Input:** Client updates $\{w_i^t\}$, previous beliefs $\{b_i^{t-1}\}$, trained DQN $Q_\phi$, validation set $\mathcal{V}$
2: **Output:** Updated global model $w^{t+1}$, updated beliefs $\{b_i^t\}$
3: **for** each client $i$ **do**
4: $\quad o_i^t \leftarrow \text{ANOMALYSIGNALS}(w_i^t)$
5: $\quad b_i^t \leftarrow \text{BAYESUPDATE}(b_i^{t-1}, o_i^t)$
6: **end for**
7: Construct state $s^t = \{(o_i^t, b_i^t)\}_{i=1}^N$
8: Select action $a^t = \arg\max_a Q_\phi(s^t, a)$
9: **for** each client $i$ **do**
10: $\quad TS^t(i) \leftarrow \text{TRUSTMAP}(b_i^t, a^t)$
11: **end for**
12: Aggregate updates:

$$w^{t+1} = \sum_{i=1}^N \frac{n_i}{n} TS^t(i) w_i^t, \qquad n = \sum_{i=1}^N n_i$$

---

### 3.3 ANOMALY EVIDENCE

Each client update is evaluated through three complementary anomaly metrics:

1. *Directional alignment:* cosine similarity between $w_i^t$ and the global model $w^t$, capturing whether an update aligns with the consensus.

2. *Magnitude deviation:* deviation of $\|w_i^t\|$ from the median update norm, highlighting abnormal scaling or sign-flip attacks.

3. *Validation impact:* the change in accuracy on a small server-side validation set when applying $w_i^t$, measuring the contribution of the update to model utility.

Individually, each signal provides only partial evidence: cosine similarity captures geometric consistency, norm deviation highlights extreme scaling, and validation impact reflects end-task performance. By concatenating them into a joint feature vector, we obtain the observation

$$o_i^t = \left(a_i^{\text{dir}}, a_i^{\text{mag}}, a_i^{\text{val}}\right),$$

which compactly summarizes client behaviour. An overall anomaly score $A(i)$ is computed as a weighted combination of these metrics, based on reputation and robust-aggregation mechanisms(Xu & Lyu, 2021).

### 3.4 BAYESIAN BELIEF TRACKING

Given observations $\{o_i^\tau\}_{\tau=1}^t$, the server maintains a belief $b_i^t$ about whether client $i$ is benign or malicious. Let $s_i^t \in \{\text{benign}, \text{malicious}\}$ denote the latent state. The posterior is updated recursively:

$$b_i^t = P(s_i^t \mid o_i^{1:t}) \propto P(o_i^t \mid s_i^t) b_i^{t-1}.$$

To prevent adversaries from escaping detection with isolated benign-looking updates, trust scores are derived from accumulated beliefs. Following the update rule introduced in the thesis, trust scores are adjusted as

$$TS^{t+1}(i) = TS^t(i) \times \left(1 - \lambda A(i) + \eta C(i)\right),$$

where $A(i)$ is the aggregated anomaly score, $C(i)$ the contribution quality, and $\lambda, \eta > 0$ hyperparameters. This ensures that evidence builds up across rounds rather than being forgotten.

### 3.5 REINFORCEMENT LEARNING DEFENCE

Beliefs provide a probabilistic summary of client trust, which we embed into a POMDP. This design follows reinforcement learning formulations for adversarial federated learning environments (Anwar & Raychowdhury, 2021; Xu & Lyu, 2021), where trade-offs between robustness and performance must be carefully optimized:

- **State:** $s^t = \{(o_i^t, b_i^t)\}_{i=1}^N$ concatenates anomaly features and beliefs.
- **Action:** adjust client weights via *increase*, *reduce*, or *hold*.
- **Reward:** A linearly weighted balance of performance, trust accuracy, cost and attack success rate,

$$R(B, a) = \alpha_{\text{perf}} \cdot \mu_P + \alpha_{\text{trust}} \cdot \tau_A - \alpha_{\text{cost}} \cdot \delta_C - \alpha_{\text{attack}} \cdot \gamma,$$

where $\mu_P$ is validation accuracy, $\tau_A$ measures correct penalization of malicious clients, and $\delta_C$ penalizes unnecessary reductions of benign ones.

A DQN (Mnih et al., 2013) agent parameterizes $Q(s, a; \theta)$ and is trained via experience replay and $\epsilon$-greedy exploration. Parameters are optimized by minimizing the temporal-difference loss

$$L(\theta) = \frac{1}{|B|} \sum_i \left[ y_i - Q(s_i, a_i; \theta) \right]^2, \quad y_i = r_i + \gamma \max_{a'} Q'(s_i', a'; \theta'),$$

where $Q'$ is the target network and $\gamma$ the discount factor.

## 4 EXPERIMENTS

We evaluate on CIFAR-10 to test robustness across attack types, participation regimes, and degrees of non-iid heterogeneity, and we ablate observability to mimic secure aggregation. Figure 2 visualises (i) test accuracy over rounds, (ii) per-client belief trajectories, and (iii) the final (mean-normalised) trust scores.

### 4.1 EXPERIMENTAL SETUP

**Datasets and architectures.** Each client trains a lightweight three-layer CNN tailored to CIFAR-10. The server hosts a DQN with two hidden layers (ReLU) that outputs Q-values over trust actions (*increase*, *reduce*, *hold*). Core hyperparameters for the agent, belief update, and reward are listed in Appendix A.1.

**Threat models.** We consider targeted backdoor attacks. For backdoors, a $k \times k$ trigger is stamped on a fraction of local batch samples and relabelled to a fixed target; attack success rate (ASR) is measured at test time.

**Non-IID partitions and participation.** Client datasets are formed with a Dirichlet prior of concentration $\alpha$ over label proportions, with a smaller $\alpha$ indicating stronger heterogeneity. We study $\alpha \in \{0.1, 0.2, \ldots, 1.0, 5.0\}$.

**Observability regimes.** To simulate partial observability (e.g., secure aggregation), we vary available anomaly signals: *Full* (directional alignment + magnitude deviation + validation impact), *No-Validation*, and *Directional-only*.

**Metrics.** We report the test accuracy over aggregation rounds, alongside the attack success rate (ASR) and ROC-AUC. Moreover, we also plot the belief-state evolution, and final trust scores to demonstrate the effectiveness of the methodology.

**Training protocol.** Unless stated otherwise: 10 clients, 50% participation, 20% malicious per round, and 50 global rounds. All experiments are in PyTorch on NVIDIA GPUs; full settings are in Appendix A.1.

### 4.2 BASELINE EXPERIMENT

**Protocol.** We present a single, end-to-end run without comparisons: 10 clients, 50% participation, 20% malicious, 50 rounds, and Dirichlet $\alpha=0.5$. The DQN uses the default configuration from Appendix A.1 (discount 0.9, $\varepsilon$-greedy with linear decay, learning rate $10^{-3}$, batch size 64).

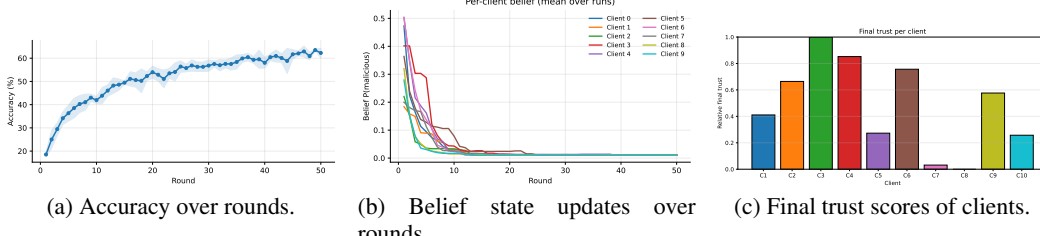

(a) Accuracy over rounds.    (b) Belief state updates over rounds.    (c) Final trust scores of clients.

Figure 2: **Baseline learning dynamics.** (a) Test accuracy over rounds rises steadily, reaching 65.32% by round 50. (b) Per-client belief trajectories separate early and then stabilize, indicating the agent's ability to track client reliability under partial observability. (c) Final trust scores at round 50 (mean-normalized) showing low weights for suspected clients while preserving high trust for consistently benign participants.

**Results.** Figure 2 summarises the main signals. Accuracy rises steadily to a final value of 65.32% by round 50. Belief trajectories separate early and then stabilise, reflecting the accumulation of directional, magnitude, and (when available) validation evidence under partial observability. The final trust bars (mean-normalised) show that belief-driven actions translate into effective aggregation by up-weighting consistently benign clients and down-weighting suspected ones. Detailed results are provided in Appendix A.3

**Takeaways.** (i) The policy converges within 50 rounds with competitive clean accuracy; (ii) belief updates provide clear separation that the DQN turns into actionable trust adjustments; and (iii) robustness emerges from sequential evidence accumulation and trust-aware weighting rather than strong single-round detectors which remains consistent with our POMDP framing.

### 4.3 Defence Controller Evaluation

**Motivation.** Choosing a control policy for defence in cross-device FL requires reasoning about non-stationarity (shifting client mixes and attacks), partial observability (limited round-level signals), long-horizon credit assignment, and tight sample budgets. We therefore compare four agent classes that span increasing representational and optimization capacity. DQN (off-policy, value-based with replay and bootstrapping) is a strong candidate for this setting: it can reuse past experience under distribution shift, stabilize learning with target networks, and approximate non-linear mappings from round features (e.g., heterogeneity indicators, anomaly signals, trust states) to actions. Policy Gradient (Williams, 1992) provides a direct, on-policy baseline that optimizes expected return without value bootstrapping but is known to exhibit higher variance and sensitivity to reward scaling which stress-tests robustness under noisy, adversarial feedback. Linear-Q (Watkins & Dayan, 1992) isolates the effect of function-approximation capacity by restricting the value function to a linear form, revealing whether simple, fast learners suffice when signals are weak or correlated. Additionally, we also compare a Random agent as a null model, establishing a lower bound on achievable control. Evaluating these agents clarifies which learning biases are most effective for reliable defence policies in federated learning.

**Results.** Under a fixed FL setting , agents exhibit markedly different learning behavior. DQN improves steadily and reaches a substantially higher final accuracy ($61.48 \pm 4.09\%$) while the remaining agents hover near chance ($\approx 16\%$). DQN also converges early ($18.4 \pm 4.0$ rounds), whereas Linear-Q, Policy Gradient, and Random show no convergence by round 50. ASR levels are comparable in magnitude but differ in stability: DQN's ASR is moderate and relatively stable ($41.35 \pm 7.67\%$), while the baselines show wide variability (e.g., Linear-Q $30.36 \pm 24.97\%$, Random $36.11 \pm 29.99\%$). ROC–AUC and ECE remain in a narrow range across agents (AUC around 0.5–0.57, ECE $\approx 0.09$). We report end-of-training test accuracy, backdoor ASR, and ROC-AUC (mean $\pm$ std) are shown in Fig. 3, and seed-wise summary statistics appear in Table 1. Further detailed results are outlined in Appendix A.3.

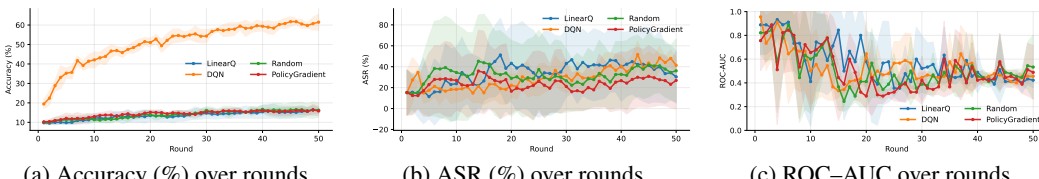

(a) Accuracy (%) over rounds.      (b) ASR (%) over rounds.      (c) ROC–AUC over rounds.

Figure 3: **Agent performance over training rounds.** We compare Linear-Q, DQN, Random, and Policy Gradient under a fixed FL setup. (a) shows test accuracy, where DQN steadily improves over rounds and attains substantially higher values than the other agents, which remain near chance. (b) shows backdoor ASR, with DQN exhibiting relatively lower and more stable levels, while the baselines fluctuate with higher variance. (c) reports ROC–AUC, where all agents trend downward with training but DQN maintains competitive performance. Overall, the results suggest that DQN achieves stronger accuracy while balancing robustness compared to simpler agents.

Table 1: Agent comparison on CIFAR-10 under a fixed FL setup (10 clients, 50% participation, 20% malicious, 50 rounds, $\alpha$=0.5). Values are mean $\pm$ std over 5 seeds. Convergence round is reported as mean only.

| Agent | Acc. (%) | ASR (%) | ROC-AUC | Convergence round | Avg. Reward |
|---|---|---|---|---|---|
| LinearQ | $16.02 \pm 1.59$ | $30.36 \pm 24.97$ | $\mathbf{0.573 \pm 0.087}$ | 50.0 | $0.584 \pm 0.151$ |
| DQN | $\mathbf{61.48 \pm 4.09}$ | $41.35 \pm 7.67$ | $0.531 \pm 0.046$ | $\mathbf{18.4}$ | $\mathbf{0.941 \pm 0.060}$ |
| Policy Gradient | $15.86 \pm 2.40$ | $\mathbf{26.88 \pm 27.30}$ | $0.506 \pm 0.075$ | 50.0 | $0.006 \pm 0.155$ |
| Random | $16.28 \pm 3.60$ | $36.11 \pm 29.99$ | $0.512 \pm 0.055$ | 50.0 | $0.005 \pm 0.110$ |

**Takeaways.** DQN is the only agent that consistently escapes chance-level performance and attains high accuracy with fast convergence, indicating advantages of off-policy value-based control with replay and bootstrapping in this non-stationary, partially observed setting. Similar end-of-round ROC–AUC and near-constant ECE across agents suggest the gap is not due to per-round separability or calibration alone, but to how the policy exploits sequential signals over rounds. Although non-DQN agents such Policy Gradient occasionally achieve very low transient ASR , these dips do not persist or translate into clean accuracy gains, underscoring the need for stable policy improvement rather than opportunistic fluctuations.

### 4.4 EFFECT OF DATA HETEROGENEITY

**Motivation.** Non-IID data is intrinsic to cross-device FL because clients collect data from different users and contexts. Heterogeneity changes both optimization dynamics and the attack surface. When client distributions overlap less, gradients drift and aggregation becomes harder, and malicious updates can gain leverage. A robust defence must therefore work across a spectrum of heterogeneity levels. The Dirichlet parameterization provides a controlled way to vary overlap through a single concentration parameter $\alpha$. Sweeping $\alpha$ allows us to test whether the policy adapts when client data move from highly skewed to more homogeneous regimes.

**Results.** Increasing $\alpha$ improves accuracy and reduces ASR. At $\alpha$=0.1, accuracy is $46.46 \pm 3.37\%$ and ASR is $61.11 \pm 17.32\%$. At $\alpha$=1.0, accuracy rises to $65.22 \pm 0.84\%$ and ASR declines to $46.73 \pm 4.28\%$. At $\alpha$=5.0, accuracy reaches $67.07 \pm 1.37\%$ and ASR falls to $44.18 \pm 4.15\%$. From $\alpha$=0.1 to $\alpha$=5.0, mean accuracy increases by 20.61 percentage points and mean ASR decreases by 16.93 percentage points. The largest gains occur when moving out of the most heterogeneous setting with $\alpha \leq 0.3$ into the moderate range with $\alpha$ between 0.4 and 1.0. Between $\alpha$=1.0 and $\alpha$=5.0 the additional gains are smaller. Accuracy improves by 1.85 points and ASR decreases by 2.55 points. We summarise means and standard deviations in Table 2

Variance contracts as $\alpha$ increases. The ASR standard deviation decreases from 17.32 at $\alpha$=0.1 to the interval 2.87 to 7.78 for $\alpha \geq 0.4$, with the minimum at $\alpha$=0.9. Accuracy variability tightens to at most 2.27 points for $\alpha \geq 0.8$. The values are 1.20, 2.27, 0.84, and 1.37 at $\alpha$=0.8, 0.9, 1.0, and 5.0. There is higher variability at some mid-range points such as 4.52 at $\alpha$=0.7 and 4.18 at $\alpha$=0.5. Auxiliary metrics remain stable across the sweep. ROC-AUC lies between 0.51 and 0.58. ECE is $0.09 \pm 0.00$ for all $\alpha$ and is therefore omitted from Table 2.

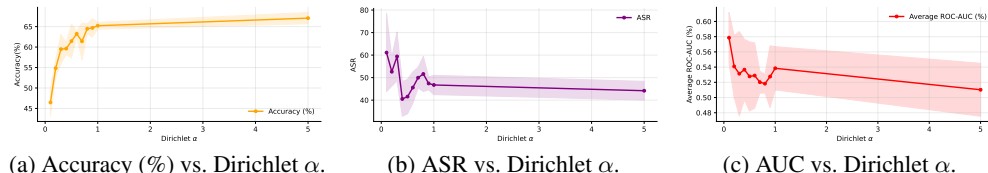

(a) Accuracy (%) vs. Dirichlet $\alpha$.    (b) ASR vs. Dirichlet $\alpha$.    (c) AUC vs. Dirichlet $\alpha$.

Figure 4: **Dirichlet heterogeneity sweep.** Increasing $\alpha$ improves clean accuracy and reduces ASR, and variability narrows as $\alpha$ grows. ROC-AUC remains largely stable across $\alpha$ with only a mild downward drift at higher $\alpha$. (a) Accuracy, (b) ASR, (c) ROC-AUC.

Table 2: Dirichlet sweep on CIFAR-10. Reported values are mean $\pm$ standard deviation over runs. ECE remains constant at $0.09$ across all $\alpha$. We observe an increase in the accuracy, and drop in ASR and ROC-AUC.

| $\alpha$ | Acc. (%) | ASR (%) | ROC-AUC |
|---|---|---|---|
| 0.1 | $46.46 \pm 3.37$ | $61.11 \pm 17.32$ | $\mathbf{0.58 \pm 0.03}$ |
| 0.2 | $54.85 \pm 2.19$ | $52.62 \pm 6.41$ | $0.54 \pm 0.04$ |
| 0.3 | $59.49 \pm 3.59$ | $59.40 \pm 10.87$ | $0.53 \pm 0.06$ |
| 0.4 | $59.62 \pm 1.43$ | $\mathbf{40.52 \pm 7.71}$ | $0.54 \pm 0.04$ |
| 0.5 | $61.44 \pm 4.18$ | $41.57 \pm 7.65$ | $0.53 \pm 0.05$ |
| 0.6 | $63.23 \pm 0.61$ | $45.58 \pm 6.98$ | $0.53 \pm 0.04$ |
| 0.7 | $61.40 \pm 4.52$ | $49.96 \pm 4.56$ | $0.52 \pm 0.01$ |
| 0.8 | $64.47 \pm 1.20$ | $51.62 \pm 7.78$ | $0.52 \pm 0.01$ |
| 0.9 | $64.68 \pm 2.27$ | $47.41 \pm 2.87$ | $0.53 \pm 0.04$ |
| 1.0 | $65.22 \pm 0.84$ | $46.73 \pm 4.28$ | $0.54 \pm 0.03$ |
| 5.0 | $\mathbf{67.07 \pm 1.37}$ | $44.18 \pm 4.15$ | $0.51 \pm 0.03$ |

**Takeaways.** Strong non-IIDness with small $\alpha$ degrades both accuracy and robustness. Increasing overlap across clients with larger $\alpha$ improves accuracy, reduces ASR, and narrows variability, with diminishing returns beyond $\alpha \approx 1.0$. ROC-AUC and ECE remain effectively constant. The robustness gains therefore arise from sequential belief updates and multi-signal fusion rather than from changes in single-round separability.

### 4.5 ANOMALY SIGNAL ABLATION

**Motivation.** Real FL systems face limited observability due to secure aggregation, privacy constraints, and bandwidth. Our defence fuses three anomaly cues—directional alignment, magnitude deviation, and (when available) validation impact—into Bayesian beliefs that drive trust-aware actions. Because these cues differ in cost and reliability (validation may be unavailable, magnitudes are noisy under non-iid data, and directional evidence is lightweight but weaker alone), ablating the signal budget (*Full*, *No-Validation*, *Directional-only*) lets us quantify robustness under realistic constraints, isolate each cue's contribution, and test whether sequential belief updates can offset weaker single-round evidence.

**Results.** Removing signals degrades robustness while leaving clean accuracy essentially unchanged. With the Full budget, the system attains $62.33\%$ accuracy, $42.84\%$ ASR, and ROC–AUC $0.55$. Dropping validation keeps accuracy flat ($62.36\%$) but increases ASR by $+2.60$ points to $45.44\%$ and lowers AUC by $0.07$ to $0.48$. Using only directional alignment yields $62.24\%$ accuracy while ASR rises to $46.96\%$ ($+4.12$ vs. Full) with AUC again at $0.48$. Overall, validation cues contribute most to detection quality and attack suppression; when they are removed, sequential belief updates still preserve clean accuracy but cannot fully offset the drop in separability.

Table 3: Anomaly signal ablation on CIFAR-10 (DQN; 10 clients, $50\%$ participation, $20\%$ malicious, 50 rounds, $\alpha=0.5$). Values are mean $\pm$ std over seeds.

| Signal budget | Acc. (%) | ASR (%) | ROC-AUC |
|---|---|---|---|
| full | $62.33 \pm 2.00$ | $\mathbf{42.84 \pm 5.86}$ | $\mathbf{0.55 \pm 0.03}$ |
| no_validation | $\mathbf{62.36 \pm 2.01}$ | $45.44 \pm 5.24$ | $0.48 \pm 0.07$ |
| directional | $62.24 \pm 2.16$ | $46.96 \pm 4.28$ | $0.48 \pm 0.01$ |

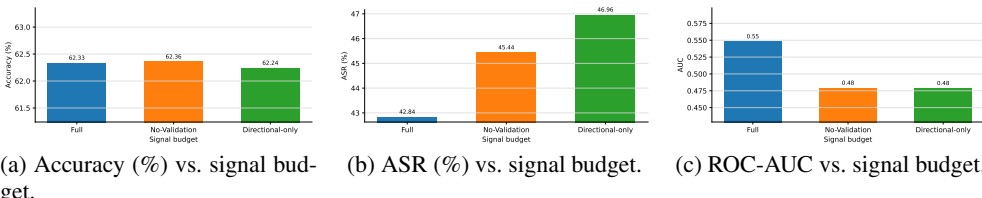

(a) Accuracy (%) vs. signal budget.

(b) ASR (%) vs. signal budget.

(c) ROC-AUC vs. signal budget.

Figure 5: **Signal-budget ablation (Full / No-Validation / Directional-only).** Clean accuracy remains nearly unchanged, while robustness degrades as evidence is removed: ASR increases and ROC-AUC drops from $0.55$ to $0.48$. Bars show means over five seeds.

**Takeaways.** Accuracy is largely invariant to the anomaly-signal budget, indicating that the control policy can preserve clean performance from round-level dynamics alone. Robustness, however, depends critically on richer signals: removing validation cues increases ASR and reduces ROC–AUC, revealing a loss of per-round separability that sequential updates cannot fully recover. Directional-only signals further degrade robustness with minimal accuracy change, underscoring that detection quality is the limiting factor. Practically, the full signal budget yields the best robustness/utility trade-off; if signals must be pruned, dropping validation incurs a smaller penalty than relying on directional alignment alone.

## 5 DISCUSSION

Our study frames federated learning defence as a sequential decision-making problem under partial observability and demonstrates the advantages of trust-aware reinforcement learning. Across baseline, heterogeneity, and signal-ablation experiments, we find that the proposed DQN agent consistently achieves higher clean accuracy while containing attack success rates. Crucially, the gains emerge not from stronger single-round anomaly detectors, but from accumulating multi-signal evidence over time and translating it into adaptive trust-weighted aggregation. This temporal reasoning enables early separation of benign and malicious clients and stabilizes trust assignments, leading to more robust global models.

Comparison against alternative controllers further highlights these dynamics. While linear approximations and policy-gradient methods occasionally achieve transient robustness, they lack the stability and long-horizon consistency of the DQN. The calibration analysis supports this view: all agents converge to similar ECE levels, suggesting that robustness gains stem from sequential exploitation of signals rather than differences in probabilistic calibration. Overall, the results provide evidence that reinforcement learning under partial observability is a principled and practical foundation for robust FL defences.

## 6 LIMITATIONS

Despite promising results, our work has several limitations. First, the experiments are constrained by GPU availability, which limited the scope of hyperparameter sweeps, model capacity, and number of seeds explored. Second, we relied on the AIJACK (Takahashi, 2024) simulator to approximate cross-device FL under secure aggregation. While this provides controlled and reproducible settings, simulators cannot fully capture the complexity of real-world deployments, including communication delays, heterogeneous hardware, or user behavior. Finally, we focused on CIFAR-10 with a lightweight CNN backbone; extending to larger vision or multimodal tasks is necessary to test scalability. Addressing these constraints is an important direction for future work before deployment in practical FL systems.

## 7 REPRODUCIBILITY STATEMENT

We provide instructions for experimental reproducibility in Appendix A.2.

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

# A    APPENDIX

## A.1    HYPERPARAMETER SETTING

Table 4: Core hyperparameters for the trust-aware DQN–POMDP defence. Defaults match our CIFAR-10 setting; ranges indicate values explored in sweeps/ablations.

| Component | Hyperparameter (symbol) | Default | Range(s) |
|---|---|---|---|
| DQN agent | Discount factor ($\gamma$) | 0.9 | $\{0.8, 0.9, 0.99\}$ |
| | Exploration start ($\epsilon_{\text{start}}$) | 1.0 | fixed |
| | Exploration end ($\epsilon_{\text{end}}$) | 0.01 | fixed |
| | Exploration decay steps | 5000 | $\{3000, 5000, 8000\}$ |
| | Replay buffer size | 10,000 | $\{5k, 10k, 50k\}$ |
| | DQN batch size | 64 | $\{32, 64, 128\}$ |
| | DQN learning rate | $10^{-3}$ | $\{10^{-4}, 10^{-3}, 10^{-2}\}$ |
| | Hidden units (per layer) | 64 | $\{32, 64, 128\}$ |
| | Target update freq. | 100 steps | $\{50, 100, 200\}$ |
| Trust & beliefs | Anomaly penalty ($\lambda$) | 0.3 | $\{0.1, 0.3, 0.5\}$ |
| | Contribution reward ($\eta$) | 0.2 | $\{0.1, 0.2, 0.3\}$ |
| | Reward: perf. weight ($\alpha$) | 1.0 | $\{0.5, 1.0, 1.5\}$ |
| | Reward: trust weight ($\beta$) | 1.0 | $\{0.5, 1.0, 1.5\}$ |
| | Reward: action cost ($\gamma_{\text{cost}}$) | 0.5 | $\{0.25, 0.5, 1.0\}$ |
| | Reward: ASR penalty ($\delta$) | 0.5 | $\{0.25, 0.5, 1.0\}$ |

**Overview.** Table 4 lists the core hyperparameters of our trust-aware DQN policy and the belief/trust update, together with the ranges explored in ablations. We keep the default configuration fixed across all reported results unless a sweep is explicitly stated. Table 5 summarizes the federated protocol, partitioning, and attack settings. (For the large Dirichlet sweep in the main paper, we extend $\alpha$ beyond the ablation ranges shown here.)

**DQN agent.** We use a light two-layer MLP with hidden width 64 to keep the control overhead small relative to client training. The replay buffer of 10k transitions and target network updates every 100 steps provide stable learning without sacrificing responsiveness. The $\varepsilon$-greedy schedule decays from 1.0 to 0.01 over 5,000 steps, which in practice yields several rounds of broad exploration before converging to near-greedy behavior. Batch size (64) and learning rate ($10^{-3}$) were chosen from small grids to balance stability and speed. Gradients are clipped (norm $\leq 1$) in all settings.

**Beliefs, trust, and reward.** Belief updates combine anomaly likelihoods with a Bayesian rule and mild smoothing; we found this prevents oscillations when signals are noisy. Trust uses multiplicative updates with an anomaly penalty $\lambda$ and a contribution reward $\eta$, both picked from short grids (Table 4). The reward weights ($\alpha, \beta, \gamma_{\text{cost}}, \delta$) trade off clean accuracy gains, action correctness,

Table 5: Experimental protocol (CIFAR-10). Defaults and sweep ranges for dataset partitioning, client training, and attacks.

| Category | Setting | Default | Range(s) |
|---|---|---|---|
| Federated setup | Total clients | 10 | $\{10, 50, 100\}$ |
| | Participating Clients / round | 5 | $\{\lfloor 0.1N \rfloor, \lfloor 0.2N \rfloor, \lfloor 0.5N \rfloor\}$ |
| | Rounds | 50 | $\{10, 15, 20, 30\}$ |
| | Dirichlet $\alpha$ (non-iid) | 0.5 | $\{0.1, 1\}$ |
| Client training | Client batch size | 32 | $\{32, 64\}$ |
| | Client LR (SGD) | 0.01 | $\{0.005, 0.01, 0.02\}$ |
| Attacks | Malicious ratio / round | 0.2 | $\{0.1, 0.2, 0.3, 0.4, 0.5\}$ |
| | Attack type | backdoor | {backdoor, sign_flip, gradient_push, collusion} |
| | Attack strength | 0.5 | $\{0.3, 0.4, 0.5, 0.6, 0.8\}$ |
| | Backdoor fraction (batch) | 0.1 | $\{0.05, 0.1, 0.2\}$ |
| | Trigger size, target label | 4, class 0 | sizes $\{3, 4, 6\}$; target $\{0 \ldots 9\}$ |

intervention cost, and the ASR change; the defaults provide a conservative defense that avoids over-filtering while reacting to rising ASR. In our ablations, moving any one weight within the shown range produced the expected monotone trade-off without destabilizing training.

**Federated protocol.** Unless otherwise noted, we use $N{=}10$ clients with $k{=}5$ participants per round and 50 global rounds. Non-IIDness is controlled via a Dirichlet prior with default $\alpha{=}0.5$; smaller values increase label skew. Malicious clients are assigned each round by vulnerability score, with a default ratio of 0.2. We evaluate multiple attacks; backdoor is the default for end-to-end runs (trigger size $4{\times}4$, batch poisoning fraction 0.1, target label 0). Collusion blends malicious updates along a shared direction.

**Tuning protocol.** We adopt a coarse-to-fine procedure: (i) select stable defaults on $\alpha{=}0.5$ with the backdoor threat; (ii) vary one block at a time (DQN, beliefs/trust, or reward) over the small ranges in Table 4; (iii) keep the best setting fixed for subsequent studies unless otherwise stated. All sweeps use identical seeds across conditions for fairness.

**Practical guidance.** If the policy under-reacts to attacks (ASR drifts up), increase $\delta$ (ASR penalty) or $\beta$ (trust weight), or slow exploration decay. If it over-prunes benign clients, reduce $\lambda$ (anomaly penalty) and/or $\gamma_{\text{cost}}$ (action cost). If Q-values oscillate, raise the target update period or buffer size, or lower the learning rate.

## A.2 Reproducibility and Seeding Protocol

**Scope.** We fix random seeds and control major sources of stochasticity (partitioning, client sub-sampling, minibatch order, and model initialization) for all reported results. Unless a sweep explicitly varies a hyperparameter, all other settings remain at their defaults (Tables 4 and 5).

**Global RNG control.** At the start of each run, a single integer seed initializes the random number generators for Python, NumPy, and PyTorch; when CUDA is available, the GPU generators on all devices are seeded as well. To avoid nondeterminism from data loading, we use a single data-loader worker so that batch shuffling depends only on the seeded generators. When exact, bitwise-identical replays are required (beyond statistical reproducibility), we enable deterministic CuDNN kernels and disable algorithm autotuning; this may modestly reduce training throughput but eliminates kernel-selection variability.

**Seed schedules by experiment.** We adopt fixed, public seed lists so that every result in the paper can be reproduced exactly. The seeds are provided in Table 6.

**Deterministic scope of seeding.** The seed $s$ deterministically controls: (i) Dirichlet partition sampling (class proportions per client), (ii) round-wise client subsampling, (iii) network weight initial-

Table 6: Seeds per experiment. For multi-condition experiments (e.g., several values of $\alpha$ or the three signal budgets), the same seed list is used independently for each condition. Reported numbers are the mean $\pm$ std over the corresponding five runs.

| Experiment | Runs per setting | Seeds used per setting |
|---|---|---|
| Dirichlet sweep (per $\alpha$) | 5 | $42 + r,\ r \in \{0, 1, 2, 3, 4\}$ (i.e., 42, 43, 44, 45, 46) |
| Baseline pipeline (end-to-end) | 5 | $\{42, 64, 128, 200, 256\}$ |
| Agent comparison (Random, Linear-Q, PG, DQN) | 5 | $42 + r,\ r \in \{0, 1, 2, 3, 4\}$ (i.e., 42, 43, 44, 45, 46) |
| Signal-budget study (Full / No-Validation / Directional-only) | 5 | $\{42, 64, 128, 200, 256\}$ |

ization (global and local models), (iv) minibatch shuffles during local training, and (v) any sampling inside attacks (e.g., randomized elements of the backdoor when used).

**Runtime environment & software versions.** All experiments were executed on NVIDIA T4 GPUs (16 GB VRAM) using PyTorch and Torchvision. Enabling deterministic CuDNN flags yields bitwise-identical repeats; without them, results remain within the reported standard deviations.

### A.3 ADDITIONAL EXPERIMENTATION RESULTS

#### A.3.1 BASELINE EXPERIMENT: ASR, ROC–AUC, AND REWARD TRAJECTORIES

**Overview.** We report per-round dynamics (mean $\pm$ std across seeds) for the baseline setting, focusing on attack success rate (ASR), detection quality (ROC–AUC), and control utility (reward). These curves complement the main-text summaries by exposing early transients, stability, and variability over aggregation rounds.

Table 7: Baseline per-round summary (every 5 rounds). Values are mean $\pm$ std across seeds.

| Round | Accuracy (%) | ASR (%) | ROC–AUC | Reward |
|---|---|---|---|---|
| 1 | $18.54 \pm 0.93$ | $1.07 \pm 2.38$ | $0.73 \pm 0.19$ | $8.45 \pm 1.35$ |
| 5 | $36.31 \pm 3.41$ | $8.90 \pm 7.67$ | $0.69 \pm 0.34$ | $2.21 \pm 2.12$ |
| 10 | $41.95 \pm 1.50$ | $20.44 \pm 10.94$ | $0.63 \pm 0.27$ | $-0.81 \pm 0.89$ |
| 15 | $49.45 \pm 0.71$ | $20.30 \pm 12.90$ | $0.38 \pm 0.12$ | $0.62 \pm 1.91$ |
| 20 | $53.96 \pm 2.66$ | $18.71 \pm 5.47$ | $0.52 \pm 0.14$ | $1.64 \pm 2.82$ |
| 25 | $56.41 \pm 2.46$ | $22.64 \pm 10.00$ | $0.48 \pm 0.03$ | $2.35 \pm 2.45$ |
| 30 | $56.82 \pm 2.46$ | $26.08 \pm 3.89$ | $0.51 \pm 0.23$ | $0.30 \pm 2.78$ |
| 35 | $58.38 \pm 3.02$ | $27.63 \pm 12.05$ | $0.47 \pm 0.05$ | $0.89 \pm 3.52$ |
| 40 | $58.03 \pm 2.28$ | $38.58 \pm 11.37$ | $0.47 \pm 0.05$ | $-1.51 \pm 4.00$ |
| 45 | $61.68 \pm 2.01$ | $42.44 \pm 11.78$ | $0.56 \pm 0.19$ | $2.81 \pm 5.49$ |
| 50 | $62.27 \pm 2.01$ | $42.89 \pm 5.85$ | $0.50 \pm 0.22$ | $-1.34 \pm 2.32$ |

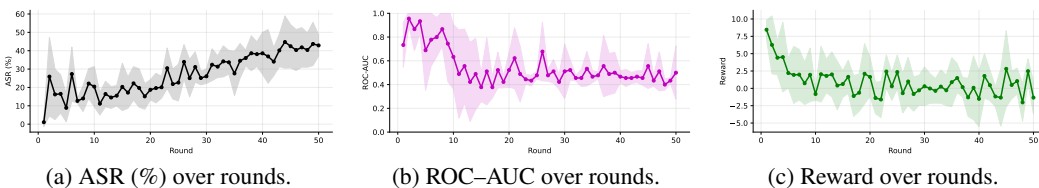

(a) ASR (%) over rounds.     (b) ROC–AUC over rounds.     (c) Reward over rounds.

Figure 6: **Baseline per-round trajectories.** Mean $\pm$ std across seeds for (a) ASR, (b) ROC–AUC, and (c) reward as training progresses over aggregation rounds.

#### A.3.2 BASELINE EXPERIMENT: TRUST-SCORE EVOLUTION

**Overview.** We summarize how the controller reweights clients over training with a single *trust-score evolution* heatmap. Rows correspond to aggregation rounds (top to bottom), columns to clients, and color intensity indicates the assigned trust at each round. A fixed color scale is used to enable direct interpretation of concentration and dispersion over time.

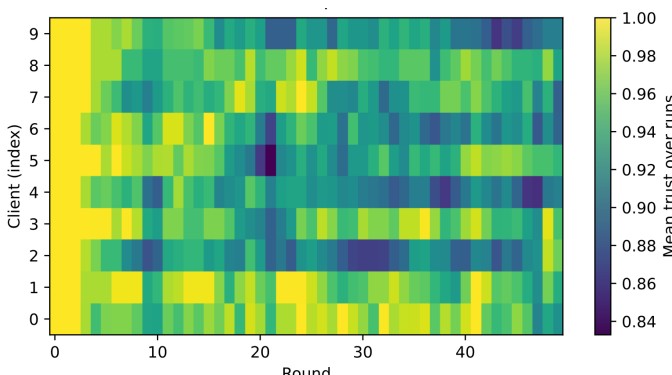

Figure 7: **Trust-score evolution heatmap.** Each row is a round and each column a client; darker cells denote higher trust. The plot reveals how trust mass shifts and stabilizes across clients as training progresses.

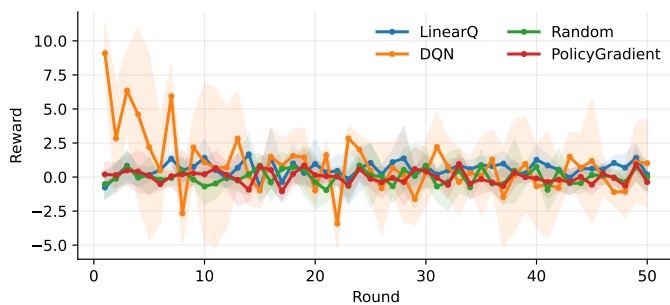

Figure 8: **Reward over rounds by agent.** Mean $\pm$ std across seeds for LinearQ, DQN, Random, and PolicyGradient under the fixed setting ($\alpha$=0.5). Higher and more stable rewards indicate faster convergence and more consistent control quality.

### A.3.3 MULTI-AGENT REWARD TRAJECTORIES

**Overview.** We report per-round reward dynamics for the defence agent comparison under the fixed FL configuration (10 clients, 50% participation, 20% malicious, 50 rounds, $\alpha$=0.5). Curves display mean $\pm$ std across 5 seeds; shaded regions denote one standard deviation.

### A.3.4 MULTI-AGENT CALIBRATION (ECE) TRAJECTORIES

**Overview.** We plot per-round expected calibration error (ECE) for the defence agent comparison under the fixed FL configuration (10 clients, 50% participation, 20% malicious, 50 rounds, $\alpha$=0.5). Curves show mean $\pm$ std over 5 seeds (shaded bands).

**Discussion.** ECE decreases sharply during the first 10–15 rounds for all agents and then plateaus around $\approx 0.09$, with minimal residual variance. DQN generally reaches the low-calibration regime slightly earlier, but the long-run calibration levels are comparable across agents. This aligns with the main-text observation that calibration is effectively constant across configurations; hence differences in accuracy and robustness arise from how policies exploit sequential signals rather than from persistent calibration gaps.

### A.3.5 TRUST-SCORE EVOLUTION ACROSS AGENTS

**Overview.** Figure 10 compares the evolution of client trust under each controller. Rows correspond to clients (ordered by final-round mean trust), columns to rounds, and colors share a common scale across panels (darker = lower trust). We average across seeds to emphasize systematic patterns rather than run-level noise.

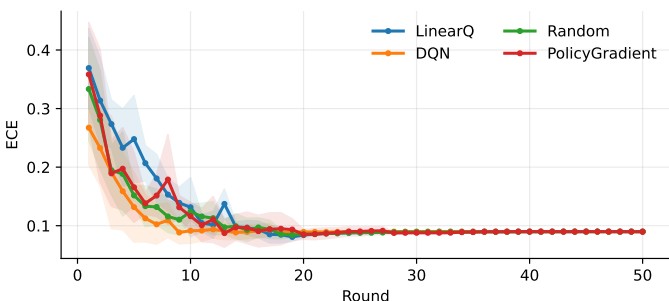

Figure 9: **ECE over rounds by agent.** Mean ± std across seeds for LinearQ, DQN, Random, and PolicyGradient at $\alpha$=0.5. All agents rapidly improve calibration in early rounds and stabilize near ECE ≈ 0.09.

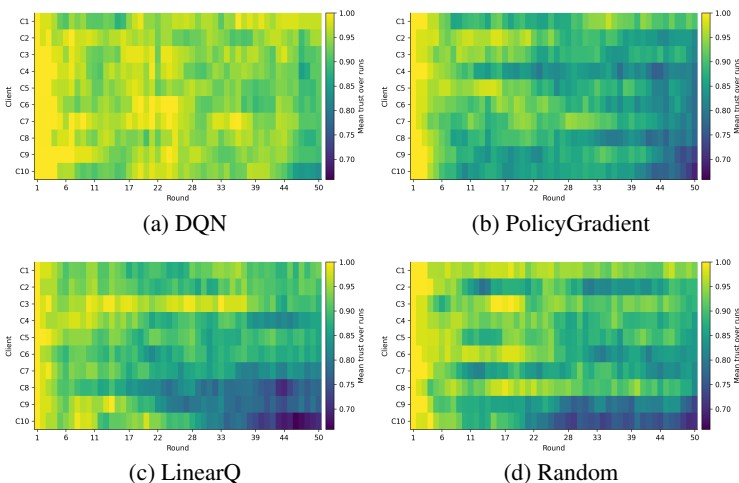

Figure 10: **Per-client trust-score evolution (mean over seeds).** Clients (rows) are ordered by final-round mean trust. DQN remains relatively uniform; PolicyGradient separates more moderately; LinearQ forms clearer high/low bands early and sustains them; Random drifts noisily with broad late-stage decline.

**Qualitative differences.** DQN exhibits early and persistent differentiation: a subset of clients is driven to lower trust within the first third of training and remains there, while high-trust clients stabilize near the upper bound. The transitions are relatively sharp, indicating decisive reweighting once sufficient evidence is accrued. Linear-Q shows near-uniform drift: trust decays slowly and broadly across clients with weak contrast between high/low groups, suggesting limited selectivity. Random yields diffuse, late decline: many clients gradually slide to lower trust with considerable temporal variability, consistent with uninformative updates. Policy Gradient sits between DQN and LinearQ: it forms moderate high/low bands but with more oscillation and slower separation than DQN.

**Takeaway.** The controllers differ primarily in how and when they separate clients. DQN concentrates trust sooner and maintains clear low-trust bands for suspected clients; the others either separate more weakly (Policy Gradient), broadly (Linear-Q), or noisily (Random). These patterns align with the per-round reward and robustness gaps reported in the main text.

A.4 LLM USAGE

We mainly utilized paraphrasing LLMs to aid in the writing process for the paper. This was to help us polish the language to help the reader in following the flow of the paper. We used ChatGPT to structure all sections of our paper and correct grammatical issues.

