# OpenReview forum: "Adaptive Federated Learning Defences via Trust-Aware Deep Q-Networks"
_ICLR.cc/2026/Conference — ICLR 2026 Conference Withdrawn Submission_

### Official Review · Reviewer_nGbF · 2025-10-28

**Soundness:** 2
**Presentation:** 2
**Contribution:** 2
**Rating:** 2
**Confidence:** 4

**Summary:**

This paper proposes a trust-aware defense mechanism for federated learning (FL), formulating the defense as a partially observable Markov decision process (POMDP). The approach leverages a Deep Q-Network (DQN) to dynamically adjust each client’s trust. Experiments on CIFAR-10 show that the DQN-based defense achieves more stable and adaptive robustness than static aggregation rules.

**Strengths:**

1. The paper reframes FL defense as sequential decision-making under uncertainty, which is a fresh and rigorous departure from static aggregation methods.
2. The integration of Bayesian belief tracking with DQN allows the system to reason temporally about client reliability is novel

**Weaknesses:**

1. Missing baseline comparisons: The experiments compare only RL controllers (DQN, Linear-Q, Policy Gradient, Random) but omit standard FL defense baselines such as FLTrust [2], FedAvg, and Krum [3]. Including these would better contextualize performance gains.

2. Limited attack evaluation: The study focuses mainly on a single backdoor attack. The proposed defense should also be tested against stronger adaptive or RL-based poisoning attacks, e.g., Li et al., NeurIPS 2022 [1].

3. Narrow experimental scale: All evaluations use CIFAR-10 with only ten clients. Results on larger or more realistic FL benchmarks and datasets would strengthen the generality claim.

[1] H. Li et al. Reinforcement Learning-based Model Poisoning Attacks on Federated Learning. NeurIPS, 2022.

[2] X. Cao et al. FLTrust: Byzantine-Robust Federated Learning via Trust Bootstrapping. arXiv:2012.13995, 2022.

[3] P. Blanchard et al. Machine Learning with Adversaries: Byzantine Tolerant Gradient Descent. NeurIPS, 2017.

**Questions:**

1. For the questions regarding experiments, please refer to the weakness part.

2. Is there any theoretical proof on the DQN convergence or performance guarantee?

3. The related-work section could be expanded to include prior trust- or reputation-based defense methods, such as Sun et al., UAI 2023 [1], Xu & Lyu 2021 [2] to better position the contribution within the literature.

[1] X. Sun et al. Pandering in a (Flexible) Representative Democracy. UAI, 2023.

[2] X. Xu and L. Lyu. A Reputation Mechanism is All You Need: Collaborative Fairness and Adversarial Robustness in Federated Learning. arXiv:2011.10464, 2021.

---

### Official Review · Reviewer_KsZM · 2025-10-31

**Soundness:** 1
**Presentation:** 1
**Contribution:** 1
**Rating:** 0
**Confidence:** 5

**Summary:**

The paper aims to improve the robustness of federated learning against Byzantine attacks. It presents a method that models the defense process as a partially observable sequential decision problem. A trust-aware Deep Q-Network controller is introduced to adjust client weights using Bayesian belief updates of client trust. The framework combines three anomaly indicators: update direction alignment, norm deviation, and validation impact. These indicators are used to estimate each client’s reliability, and the aggregation weights are then updated based on these trust estimates.

**Strengths:**

+ The idea of viewing federated defense as a partially observable reinforcement learning problem is conceptually interesting. It connects adaptive control and security under uncertainty.
+ The proposed workflow of signal extraction, trust estimation, aggregation, and reward design is clear and easy to follow.

**Weaknesses:**

- The reported attack success rates remain high (about 40–47%), and the ROC-AUC values are close to random (around 0.5). These results show that the proposed method does not achieve real robustness improvements. The final test accuracy on CIFAR-10 (around 65%) is also low compared with standard federated learning benchmarks, which limits the practical contribution.
- The comparison omits standard robust aggregation methods such as Krum, Trimmed Mean and Coordinate-wise Median, FLTrust, and FLARE . The paper only compares several reinforcement learning controllers. Without these baselines, the claims of superiority are not well supported.
- The method depends on per-client alignment and validation-impact signals. These signals require access to individual client updates, which is not possible under secure aggregation protocols. This assumption makes the method difficult to apply in real federated learning systems where client updates are encrypted or aggregated securely.
- Important components such as the observation likelihoods $P(o_t^i|s_t^i)$, prior distributions, and the trust update rule are only described in general terms. The paper does not include complete definitions or derivations. The Bayesian belief update cannot be reproduced based on the current description, which reduces transparency and technical rigor.
- The experiments are conducted with a small convolutional model on CIFAR-10 using 10 clients, 50 communication rounds, and 20 percent malicious clients. This setup is too small to represent real federated learning scenarios that involve large numbers of clients and heterogeneous data. The paper also evaluates only one type of backdoor attack. Other important attack types such as gradient sign, label flipping, and collusion are not tested.
- The Linear-Q and Policy Gradient baselines perform very poorly. This indicates weak or inconsistent hyperparameter tuning. The imbalance in training makes the comparison unfair and overstates the benefits of the DQN controller.
- The appendix states that ChatGPT was used to organize and write all sections. The style of writing, with repetitive phrasing and generic explanations, supports this statement. Although the use of language models for assistance is not unethical, the dependence on generated text appears to have replaced original explanation and analysis. The theoretical parts are shallow, and some equations seem copied from previous works without clear reference.
- All cited papers are real, but many entries contain incorrect information. Several key works are assigned the wrong year or venue. For example, FedAvg was introduced in 2017, not 2023; Yin et al. should refer to ICML 2018, not 2021; and Xu & Lyu should be listed as 2020, not 2021. Such citation errors indicate a lack of careful verification and reduce the credibility of the work.

**Questions:**

1.How are the per-client alignment and validation-impact signals computed under secure aggregation? If raw updates are used, what are the privacy and security implications?

2.What are the explicit mathematical forms of the observation likelihoods and priors used in the belief-update process?

3.How is the trust-accuracy term ($\tau_A$) in the reward defined without access to ground-truth labels for malicious clients?

4.Why are standard robust aggregation baselines such as Krum, Trimmed Mean, FLTrust, and FLARE not included in the experiments?

5.What hyperparameter tuning strategy was used for the reinforcement learning baselines to ensure fair comparison?

6.Can the controller scale to larger numbers of clients? Please include the computational complexity and runtime analysis.

---

### Official Review · Reviewer_Vgrh · 2025-11-01

**Soundness:** 3
**Presentation:** 3
**Contribution:** 3
**Rating:** 6
**Confidence:** 2

**Summary:**

This paper reformulated FL defense as a partially observable Markov decision process. Then it introduced a trust-aware Deep Q-Network that learns to optimize robustness-accuracy trade-offs by adaptively filtering and reweighting client updates. Experimental results validated the effectiveness of the proposed framework.

**Strengths:**

(1) It reformulated FL defense as a partially observable Markov decision process.

(2) A trust-aware Deep Q-Network was proposed to optimize robustness-accuracy trade-offs by adaptively filtering and reweighting client updates.

(3) Experimental results validated the effectiveness of the proposed framework.

**Weaknesses:**

(1) The experiment results are weak. More FL datasets and large numbers of clients can be included to validate the effectiveness of the proposed framework in various settings.

(2) More recent FL defense baselines, such as Byzantine-robust aggregation, can be used in the experiments.

(3) It is unclear why DQN enables faster convergence than Linear-Q, Policy Gradient, and Random.

**Questions:**

(1) What is the "update rule introduced in the thesis" in line 205?

---

### Official Review · Reviewer_dp8Z · 2025-11-01

**Soundness:** 3
**Presentation:** 3
**Contribution:** 2
**Rating:** 4
**Confidence:** 5

**Summary:**

This paper design a defense mechanism against poisoning and backdoor attacks in FL, particularly addressing the challenge of partial observability introduced by secure aggregation protocols. The core idea is to model the defense as a Partially Observable Markov Decision Process (POMDP) and use a Trust-Aware Deep Q-Network (DQN) to learn a dynamic aggregation policy that optimizes a long-horizon robustness-accuracy objective.

**Strengths:**

1. The DQN controller achieves the best robustness-accuracy trade-off compared to competitive baselines, including Random, Linear-Q, and Policy Gradient controllers.
2. The sequential belief updates are shown to effectively mitigate the weaknesses of individual, weaker anomaly signals.

**Weaknesses:**

1. The DQN-based approach introduces additional complexity and training overhead compared to static aggregation rules, which may be a concern in resource-constrained FL settings.
2. All experiments are conducted on CIFAR-10, performance on more complex or domain-specific datasets (e.g., medical or NLP tasks) remains unverified.
3. The DQN-based approach needs massive rounds to get covergence so that can get a robust aggregation policy. The communication round 50 might not sufficient, or the task is too simple so that only needs few rounds.

**Questions:**

See weakness.

---

### Note · Authors · 2025-11-12

I have read and agree with the venue's withdrawal policy on behalf of myself and my co-authors.